# A Wnt-BMP4 Signaling Axis Induces MSX and NOTCH Proteins and Promotes Growth Suppression and Differentiation in Neuroblastoma

**DOI:** 10.3390/cells9030783

**Published:** 2020-03-23

**Authors:** Marianna Szemes, Zsombor Melegh, Jacob Bellamy, Alexander Greenhough, Madhu Kollareddy, Daniel Catchpoole, Karim Malik

**Affiliations:** 1Cancer Epigenetics Laboratory, School of Cellular and Molecular Medicine, University of Bristol, Bristol BS8 1TD, UK; jb17872@bristol.ac.uk (J.B.); pazag@bristol.ac.uk (A.G.); madhu.kollareddy@bristol.ac.uk (M.K.); 2Department of Cellular Pathology, Southmead Hospital, Bristol BS10 5NB, UK; Zsombor.Melegh@nbt.nhs.uk; 3The Kids Research Institute, The Children’s Hospital at Westmead, Westmead, New South Wales 2145, Australia; daniel.catchpoole@health.nsw.gov.au

**Keywords:** neuroblastoma, BMP4, Wnt and Notch signaling, RNA sequencing, growth inhibition

## Abstract

The Wnt and bone morphogenetic protein (BMP) signaling pathways are known to be crucial in the development of neural crest lineages, including the sympathetic nervous system. Surprisingly, their role in paediatric neuroblastoma, the prototypic tumor arising from this lineage, remains relatively uncharacterised. We previously demonstrated that Wnt/β-catenin signaling can have cell-type-specific effects on neuroblastoma phenotypes, including growth inhibition and differentiation, and that BMP4 mRNA and protein were induced by Wnt3a/Rspo2. In this study, we characterised the phenotypic effects of BMP4 on neuroblastoma cells, demonstrating convergent induction of MSX homeobox transcription factors by Wnt and BMP4 signaling and BMP4-induced growth suppression and differentiation. An immunohistochemical analysis of BMP4 expression in primary neuroblastomas confirms a striking absence of BMP4 in poorly differentiated tumors, in contrast to a high expression in ganglion cells. These results are consistent with a tumor suppressive role for BMP4 in neuroblastoma. RNA sequencing following BMP4 treatment revealed induction of Notch signaling, verified by increases of Notch3 and Hes1 proteins. Together, our data demonstrate, for the first time, Wnt-BMP-Notch signaling crosstalk associated with growth suppression of neuroblastoma.

## 1. Introduction

The canonical Wnt signaling pathway is a critical regulator of differentiation, proliferation, stemness, and determination of cell fate. Deregulation of Wnt signaling contributes to many cancers, resulting from both activating mutations of the proto-oncogene *CTNNB1* (encoding β-catenin), and loss of function mutations in negative regulators of the pathway, such as *APC* (encoding Adenomatous Polyposis Coli). These changes result in elevated nuclear β-catenin, which acts a co-activator for TCF/LEF transcription factors and a pro-growth/survival gene expression programme, exemplified by transcriptional activation of oncogenes such as *MYC* and *CCND1* [1,2]. Wnt signaling in cancer can display extensive crosstalk with other morphogenetic pathways such as bone morphogenetic protein (BMP) and Notch signaling [3].

Given the developmental origins of the childhood cancer neuroblastoma, it was reasonable to examine whether deregulated Wnt signaling might be one of its central oncogenic drivers. Neuroblastoma is derived from the sympathoadrenal lineage of the neural crest [4,5], and neural crest cell (NCC) fates are contingent on tightly regulated orchestration of many signaling pathways prompting neural crest induction and specification, including Wnt/β-catenin [6,7] and BMP signaling [8]. Additionally, a well characterized differentiation block of NCCs, resulting in neuroblastoma, depends on *MYCN* amplification and over-expression [9], with MYCN transcriptionally repressing genes required for sympathetic nervous system differentiation [10,11]. In other developmental contexts, MYCN is known to be a Wnt-induced gene [12], circumstantially supporting the possible oncogenicity of Wnt signaling in neuroblastoma.

Our previous work demonstrated high-levels of the Leucine Rich Repeat Containing G Protein-Coupled Receptor 5 (LGR5) mRNA and protein in undifferentiated neuroblastomas and neuroblastoma cell-lines and that LGR5 was also an upstream regulator of Mitogen-Activated Protein Kinase (MAPK) signaling in neuroblastoma [13]. LGR5 canonically functions as an R-Spondin receptor and increases Wnt/β-catenin signaling amplitude [14]. However, we found that Wnt3a/Rspo2 treatment of neuroblastoma cell-lines did not lead to the induction of *MYCN*, in fact MYCN and MYC protein levels decreased with Wnt3a/Rspo2 treatment, in contrast to previous reports suggesting induction of *MYC* in non-MYCN amplified (non-MNA) neuroblastomas due to Wnt/β-catenin signaling [15]. Further phenotypic analysis of Wnt3a/Rspo2 treated neuroblastoma cell-lines revealed that Wnt/β-catenin signaling exerted context-dependent effects, including the growth suppression and differentiation evident in SK-N-BE(2)-C and SH-SY5Y cells. In order to understand the Wnt-induced phenotypic changes, we conducted RNA sequencing and identified 90 high-confidence Wnt/β-catenin signaling target genes in SK-N-BE(2)-C cells. A bioinformatic analysis of these neuroblastoma Wnt target genes in primary tumor datasets showed that the 90 genes contained four distinct Wnt gene modules, or metagenes, the expression of which correlated with prognosis. Wnt metagenes 1 and 2, containing approximately 56% of our neuroblastoma Wnt target genes were expressed at markedly lower levels in high-risk neuroblastomas suggesting that these genes likely encode growth-suppressive and/or pro-differentiation proteins [16]. Consistently with this idea, some genes with documented tumor-suppressive roles in neuroblastoma were included in these Wnt modules. These include *EPAS1* [17] and *MSX1* [18], both of which have been shown to inhibit neuroblastoma cell growth. Further analysis of our Wnt differentially expressed genes (DEGs) suggested that Wnt signaling is also a key regulator of mesenchymal and adrenergic differentiation states [19] which contribute to neuroblastoma cellular heterogeneity in both cell-lines and primary tumors [20,21].

One of our most highly Wnt-induced genes following Wnt induction was *BMP4*. Previous studies have alluded to a role for BMPs in neuroblastoma growth and differentiation, including BMP2 in mouse neuroblastoma and SH-SY5Y cells [22,23] and BMP4, which also affected SH-SY5Y differentiation and decreased proliferation markers [24]. However, the mechanisms and signaling crosstalk involved in BMP-mediated phenotypes and the significance to primary disease of BMPs is not known. Given the interplay of Wnt and BMP signaling in neural crest development [25,26] and in many cancers [27,28,29], we hypothesised that the Wnt-BMP pathway may be key in regulating neuroblastoma growth. Specifically, although BMPs can have context-dependent roles in cancer, either promoting or inhibiting growth [30], Wnt-BMP signaling may be at the core of a growth-restrictive module in neuroblastoma, possibly via convergence on MSX transcription factors, which are known to be downstream of BMPs in neural crest specification [31] and also inhibit the growth of neuroblastoma cells [18]. 

In this study, we examined the relationship between Wnt and BMP signaling in neuroblastoma using functional assays and next-generation sequencing. Our study establishes co-ordinate signaling pathways operational in neuroblastoma which may be exploited for prognosis and therapeutics.

## 2. Materials and Methods

### 2.1. Neuroblastoma Cell Lines and Culture Conditions

Neuroblastoma cell lines were purchased from the European Collection of Authenticated Cell Cultures (ECACC) and from Deutsche Sammlung von Mikroorganismen und Zellkulturen (DSMZ). The identity of SK-N-BE(2)-C, SH-SY5Y and IMR32 cell lines was verified by using STR profiling (Eurofins) and lack of Mycoplasma infection was confirmed by a Mycoalert Mycoplasma Detection Kit (Lonza). Cell lines were cultured in Dulbecco’s modified eagle’s medium (DMEM):F12-HAM (Sigma) supplemented with 10% (*v/v*) foetal bovine serum (FBS) (Life technologies), 2 mM L-glutamine, 100 U/mL penicillin, 0.1 mg/mL streptomycin, and 1% (*v/v*) non-essential amino acids. BMP4 (R&D Systems) treatment was carried out at low serum conditions (1–5% FBS) in the same growth media.

### 2.2. Incucyte Live Cell Imaging and Cell Cycle Analysis

Cell proliferation was monitored real time by using Incucyte Live Cell Imaging system using cell confluence as surrogate for growth, according to the manufacturer’s instructions. Briefly, cell confluence was measured in triplicate or quadruplicate wells, normalized for starting confluence, and the average normalized confluence was plotted in function of time. The statistical significance of normalized confluence between treated and control wells was assessed by using T tests for each time point. For the analysis of neuritogenesis, a digital mask was created from training image sets to enable the software to correctly recognize neurites. Neurite lengths were recorded per mm^2^ area in replicate wells, normalized to initial values and assessed for significance using T tests. 

Propidium-iodide labelling and fluorescence activated cell sorting (FACS) analysis to detect cell cycle phases was performed as previously described [32]. Briefly, floating and adherent cells were collected, washed with PBS and subsequently fixed with ice cold 70% (*v/v*) ethanol. After washing with PBS, the cells were treated with RNase A (Qiagen). After adding 50 µg/mL Propidium Iodide (Sigma), the samples were incubated at 37 °C for 15 minutes and analysed using Fluorescence Activated Cell Sorter LSRFortessa^TM^ X-20 (BD Biosciences). At least 15,000 events were collected for every replicate. The data obtained were analyzed using FlowJo software. 

### 2.3. Protein Extraction and Western blot

Cells were lysed in Radioimmunoprecipitation assay (RIPA) buffer and protein concentration was determined by using a Micro BCA TM protein assay kit (Thermo Fisher). Ten to twenty-five μg protein was loaded onto 8–10% SDS poly-acrylamide gels and run in 1× Tris-glycine SDS buffer. The proteins were transferred onto PVDF membrane (Millipore) by using wet transfer (Bio-Rad). The membrane was blocked in 5% (*w/v*) skimmed milk in PBS, rinsed and incubated with primary antibody solution at 4 °C overnight with rotation. After three washes in PBS, the membrane was incubated with HRP labelled secondary antibody solution with agitation. After further washes, the membrane was placed in ECL reagent (Seracare) and subsequently exposed to X-ray film. The antibodies used are listed in Appendix A. Western blot image data were quantified by using ImageJ software. The target protein band density was normalized to the respective loading control band (β-actin) and subsequently to the normalized intensity of the untreated sample.

### 2.4. RNA Extraction, Reverse Transcription and qPCR

RNA was extracted by using the miRNeasy kit (QIAGEN). Cells were lysed in 350 μL Qiazol reagent and vortexed to homogenize the sample. After adding 70 μL chloroform, the samples were mixed and spun at 12,000× *g*, 4 °C for 15 minutes. The supernatant was carefully removed, mixed with 100% ethanol and loaded onto RNeasy mini columns. After a wash in RWT buffer, on column DNase treatment was performed using RNase-free DNase (Qiagen). After further washes with RWT and RPE buffers, RNA was eluted in RNase-free water. Concentration and quality were assessed using a Nanodrop spectrophotometer. One μg RNA was transcribed with Superscript IV (Invitrogen) using a mixture of oligodT and random hexamer primers, according to the manufacturer’s instructions. Quantitative PCR was performed by using QuantiNova kit on Mx3500P (Stratagene). An assay for the house-keeping gene *TBP* was used as a normalizing control. Relative gene expression was calculated using the ΔΔCt method – log2 fold changes between treated and control samples were calculated after normalization to TBP. The statistical significance of log-transformed changes in gene expression was evaluated by using T tests. The oligonucleotide primers used in this study are shown in Appendix A. 

### 2.5. Immunohistochemistry

Tissue microarrays (TMAs), containing 47 pre-chemotherapy, peripheral neuroblastic tumors were stained by using a BMP4 antibody (EPR6211, Abcam). Immunohistochemistry staining was scored as positive or negative by a pathologist blinded to the specimens. All human tissues were acquired in compliance with the NSW Human Tissue and Anatomy Legislation Amendment Act 2003 (Australia). Ethics clearances 09/CHW/159 and LNR/14/SCHN/392 were approved by the Sydney Children’s Hospital Network Human Research Ethics Committee to construct TMAs and use clinical data, which was de-identified. Immunohistochemistry was performed with a Leica Microsystem Bond III automated machine using the Bond Polymer Refine Detection Kit (Ref DS9800) followed by Bond Dab Enhancer (AR9432). The slides were dewaxed with Bond Dewax Solution (AR9222). Heat mediated antigen retrieval was performed using Bond Epitope Retrieval Solution for 20 min. 

### 2.6. RNA-seq and Bioinformatic Analysis

IMR32 cells were treated with 5 ng/mL BMP4 and 2% (*w/v*) BSA (PBS) vehicle as control for 24 h and were subsequently harvested. RNA was extracted by using a miRNeasy Mini Kit (Qiagen) as described above. RNA concentration and quality were checked by using a Nanodrop spectrophotometer and a bioanalyzer (Agilent). All RNA integrity values (RIN) were above 9. cDNA libraries were prepared from 1 μg RNA as template, using the TruSeq Stranded Total RNA Library Prep Kit (Illumina) according to the manufacturer’s instructions. The libraries were sequenced by using the paired-end option with 100 bp reads on Illumina HiSeq2000 and min. 50 million reads were obtained per sample. The reads were aligned to the human genome (hg38) by using STAR and the alignment files (BAM) files were further analysed in SeqMonk v1.45. (https://www.bioinformatics.babraham.ac.uk/projects/seqmonk/). Gene expression was quantified by using the Seqmonk RNA-seq analysis pipeline. Differentially expressed genes (DEG) were identified by DESEQ2 (*p* < 0.005), and a minimum fold difference threshold of 1.3 was applied. RNA sequencing data is available from the European Nucleotide Archive (ENA) under the study accession number PRJEB36530. We performed Gene Signature Enrichment Analyses (GSEAs) on preranked lists of log2-transformed relative gene expression values (Broad Institute). Kaplan Meier survival analysis and K means clustering were performed by using the R2 Genomics Analysis and Visualization Platform (http://r2.amc.nl).

### 2.7. Statistical Analysis

Statistical analysis of quantitative PCR data was performed on log-transformed fold change values, by using T tests. Gene Set Enrichment Analysis of RNA-seq data was evaluated based on Normalised Enrichment Score (NES) and False Discovery Rate (FDR), which was calculated based on permutation of genes with a rank score. Normalized confluence of cells, which was used as a surrogate for growth, was recorded at regular intervals in replicate samples and evaluated for significant differences at each time point by using T tests. Normalized neurite length, identified by Incucyte Live Cell Imaging System, was also assessed for statistical significance with T tests based on three measurements in separate wells. These experiments have been done three times (*n* = 3). Association of differentiation status and patient survival with BMP4 expression in neuroblastic tumours, evaluated by using IHC on tissue microarrays, was assessed with Chi-squared tests. Differentially expressed genes (DEGs) in BMP4-treated neuroblastoma cells, detected by using RNA-seq, was assessed using the statistical model implemented in DESEQ2. Overlap of Wnt and BMP4-induced DEGs in neuroblastoma cells was queried for significance by using a hypergeometric test. The significance of correlation of gene or metagene expression with overall survival probability, as evaluated using a Kaplan–Meier survival analysis, was performed with log rank tests. Significance of correlation between K mean clustering of SEQC NB data set according to Wnt and BMP4-regulated genes was assessed using hypergeometric test. Association of clinical correlates with BMP4-regulated genes in TARGET-NBL data set was evaluated by using ANOVA. 

## 3. Results

### 3.1. Cross-Talk between Wnt and BMP Signaling in Neuroblastoma

In our RNA-seq analysis of Wnt3a/Rspo2-treated neuroblastoma cells, BMP4 was amongst the most highly induced genes (>50-fold) and BMP4 protein was also upregulated [16], leading us to further analyse global effects of Wnt3a/Rspo2-treatment on BMP/TGF gene sets. As shown in Figure 1A, a volcano plot of Gene Set Enrichment Analysis (GSEA) highlights the activation of all BMP/TGF-related functional gene modules in comparison to the C2 collection of gene sets in the Molecular Signatures Database (MSigDB, Broad Institute). We verified several receptors and ligands included in these gene sets by qRT-PCR. Although BMP4 was the most highly induced ligand gene, BMP2, BMP6 and BMP7 also showed 2–3 fold induction (Figure 1B). 

MSX1 and MSX2, which are known to be downstream of BMP4 [31], were also clearly induced, prompting us to assess the effects of Wnt3a/Rspo2 and BMP4 treatments at the protein level, using an antibody that recognises both MSX proteins. As seen in Figure 1C, analysis of treatments of 3 neuroblastoma cell lines consistently demonstrated that both Wnt and BMP4 ligands were able to strongly induce the MSX transcription factors, albeit to different degrees and with some selectivity of the paralogues induced apparent. Consistently with the convergence of Wnt and BMP signaling on MSX proteins, GSEA analysis confirmed a profound effect of Wnt3a/Rspo2 on the MSX1-regulated neuroblastoma transcriptome (Figure 1D).

Taken together, our data support Wnt and BMP4 signaling co-operating to regulate the neuroblastoma transcriptome, at least in part by convergence on MSX induction. 

### 3.2. BMP4 Signaling Affects Growth and Differentiation of Neuroblastoma Cells

We had previously demonstrated that Wnt ligands could inhibit the growth of neuroblastoma cells, and also influence their differentiation state [16,19]. Given the intersection of Wnt and BMP signaling suggested by our transcriptomic data, we next sought to directly examine the phenotype of neuroblastoma cell-lines treated with BMP4. SK-N-BE(2)-C cells treated with as little as 0.1 ng/mL BMP4 exhibited markedly decreased proliferation, with concentrations ranging between 1 ng/mL and 50 ng/mL essentially arresting cell-growth (Figure 2A–C). Activation of the BMP/TGF pathway was confirmed by the robust elevation of phospho-SMAD1/5/9. There was a marked increase of Tropomyosin receptor kinase A (TrkA), a well established marker of good prognosis in neuroblastoma [33]. Although immunoblotting with cleaved caspase 3 antibody showed that there was no apoptosis, an increase of the cell-cycle inhibitor p27 was evident, together with a decrease of MYCN and E2F1 proteins (Figure 2D, Appendix A). These markers are indicative of G1/S-phase cell cycle arrest, and cell-cycle analysis also showed a significant increase of cell population in G1 (Appendix A). 

We next assessed a second MYCN-amplified (MNA) neuroblastoma cell-line, IMR32, confirming the growth-inhibitory effects of BMP4, with the lowest significant effect observed at 5 ng/mL (Figure 3). In general, growth suppression was not as marked in IMR32 cells relative to SK-N-BE(2)-C cells; however, we observed clear neuritogenesis at concentrations of 1ng/ml and above (Figure 3B,D). A protein analysis again confirmed a robust phosphorylation of SMAD1/5/9, a lack of apoptosis, and induction of p27 and TrkA (Figure 3E, Appendix A). E2F1 was again decreased, but MYCN levels were not markedly affected. Consistently with the increase of neurites, dopamine β-hydroxylase (DBH) protein expression was induced by BMP4 treatment. In order to evaluate whether the effects of BMP4 were restricted to MNA neuroblastoma only, we also assessed the non-MNA neuroblastoma cell-line SH-SY5Y, and found that, similar to SK-N-BE(2)-C cells, BMP4 induced growth suppression. Like SK-N-BE(2)-C, SH-SY5Y cells showed no overt signs of neuritogenesis (Appendix A).

Together, these phenotypic analyses show that BMP signaling can block the proliferation of neuroblastoma cells, as well as inducing differentiation, similar to our findings with Wnt signaling in neuroblastoma [16].

### 3.3. BMP4 Protein Expression Correlates with Better Prognosis of Neuroblastoma Patients

It is well known that BMP signaling can have both pro- and anti-proliferative effects, depending on cell context [30]. Having established that BMP4 can restrict neuroblastoma proliferation and induce differentiation, we next sought to establish whether BMP4 expression in primary tumors would also reflect a potential tumor-suppressive function. Immunohistochemistry on neuroblastoma tissue-microarrays (TMAs) containing 47 neuroblastic tumor patient cores of different stages revealed that BMP4 expression was markedly restricted to tumors with more differentiation (ganglioneuroblastoma and ganglioneuroma), with virtually no expression of BMP4 evident in poorly differentiated neuroblastomas, including both MYCN amplified and non-amplified tumors (Figure 4A–C). Positive and negative controls for immunostaining are shown in Appendix A. Statistical analysis confirmed the marked association of BMP expression with level of differentiation (*p* = 0.0002, Figure 4D). Consistent with this, decreased BMP4 mRNA in the transgenic Th-MYCN-driven mouse neuroblastoma model, relative to normal ganglia, was also revealed by analysis of a published dataset (Appendix A) [34]. From the survival data available we could also ascertain a significant association (*p* = 0.013) of low BMP4 expression and poor survival (Figure 4E). Thus, the BMP4 expression pattern in primary tumors aligns with our in vitro functional analyses, strongly suggesting a pro-differentiation and anti-growth role for BMP signaling, particularly BMP4, in neuroblastoma. 

### 3.4. Wnt and BMP4 Signaling Have Overlapping but Distinct Effects on the Neuroblastoma Transcriptome

In order to better define the genes and pathways affected by Wnt and BMP pathways in neuroblastoma cells, we conducted RNA sequencing of IMR32 cells treated with BMP4. We identified 772 upregulated and 559 down-regulated genes (collectively referred to as BMP4 DEGs) after 24 h of BMP4 treatment. Known targets of BMP signaling such as *NOG*, *GREM2*, *GREM1* and *BAMBI* were strongly induced, together with the ID family of transcription factors, further verifying a functional, canonical BMP signaling pathway in IMR32 cells. Several Notch pathway genes were strongly induced, including *NOTCH3*, *HES1* and *HEY1* (Figure 5A). We found a highly significant overlap (*p* < 0.001) of 31 genes between the 90 high confidence Wnt-regulated genes we identified previously in SK-N-BE(2)-C cells [16] and the 1331 BMP4-regulated genes (*p* ≤ 0.005, DESEQ2 test, min. fold change 1.3), although 11 genes were oppositely regulated The complex interplay of BMP and Wnt signaling is also demonstrated by the strong but opposite regulation of the non-canonical Wnt ligand *WNT11* (Figure 5B).

In order to confirm that the IMR32 BMP4 regulated genes are representative of neuroblastoma generally, we validated a panel of 5 upregulated and 5 downregulated BMP4 targets in SK-N-BE(2)-C and IMR32 cells following BMP4 treatment. Strong concordance in BMP4 response was apparent between both cell lines (Figure 5C). Interestingly, the epigenetically silenced neuroblastoma tumor suppressor gene *CLU* was induced in both cell lines, together with another tumor suppressor *CASZ1* only in IMR32 cells [35,36]. GSEA revealed that BMP4 induced strong upregulation of Wnt pathway genes in IMR32, as well as downregulation of E2F1 and DNA replication signatures (Figure 5D). We also constructed a gene set based on a functional neuroblastoma-specific MYCN signature of 157 genes whose up- or down-regulation in IMR32 cells had been demonstrated to be strongly linked with neuroblastoma prognosis (MYCN157) [37]. BMP4 treatment had a repressive effect on the MYCN-activated members of the MYCN157 signature and also negatively regulated pediatric cancer markers. Upregulation of the Notch signaling pathway genes was confirmed by GSEA (Figure 5D). In order to examine a potential crosstalk of signaling pathways at a global level, we plotted normalized enrichment scores (NES) of GSEAs for our previous Wnt RNA-seq together with our BMP4 RNA-seq data, using the C2 collection of Molecular Signatures Database (Broad Institute). We found not only a strong mutual upregulation of genes participating in Wnt and BMP signaling, but also Notch signaling with both BMP and Wnt (Figure 5E).

We previously utilised our Wnt DEGs for k-means clustering of RNA-seq expression data of primary neuroblastoma tumors (SEQC, GSE62564 [38]) to demonstrate that they are capable of segregating clinical subtypes of neuroblastoma, such that a Wnt DEGs-dependent low risk (Wnt-LR), intermediate risk (Wnt-IR), and two high-risk (with or without MNA,Wnt-HR) categories could be identified [16]. We therefore tested whether the expression patterns of our BMP4 DEGs, which included both up and downregulated genes, in primary tumors might also be able to partition clinical subtypes and be predictive of outcome. As shown in Figure 6A, k-means clustering of RNA-seq data of 498 primary neuroblastomas (SEQC) with the 1331 BMP4 DEGs identified 3 patient clusters, corresponding to 3 distinct risk categories. Kaplan-Meier curves for the three BMP clusters demonstrated that BMP4 cluster 1 was associated with low-risk, BMP4 cluster 2 with intermediate risk, and BMP4 cluster 3 with high risk (Figure 6B). Further, BMP4 cluster 1 aligned remarkably closely with Wnt-LR, BMP4 cluster 2 with Wnt-IR, and BMP4 cluster 3 with Wnt-HR (*p* = 1.1e–150 Chi-squared test, Appendix A). Kaplan–Meier analysis of BMP4 upregulated genes showed that their high expression correlates with good prognosis in the SEQC patient set, whereas high expression of BMP4 downregulated genes correlates with poor prognosis (Figure 6C–D), suggesting that BMP4 induces gene expression programmes associated with less aggressive clinical subtypes. Similarly, expression of neuroblastoma specific BMP4 upregulated genes significantly correlated with better survival, favourable histology and lower expression of Ki67 proliferation marker in another NB tumour data set, TARGET-NBL (Appendix A). Conversely, BMP4 downregulated genes showed association with poor prognosis and unfavourable histology and high Ki67 immunoreactivity. 

Taken together, our RNA-seq reveals profound effects of BMP4 on the neuroblastoma cell transcriptome, including interaction with Wnt and Notch signaling. Importantly, our analyses strongly support a growth-suppressive function for BMP4 signaling in neuroblastoma.

### 3.5. Notch Signaling Is Downstream of Wnt-BMP Signaling

Our analysis of Wnt DEGs has suggested interactions between Wnt and Notch signaling in neuroblastoma [16]. As MSX1 has been shown to upregulate Notch signaling [18], we next examined the links between Wnt, BMP4, MSX1 and Notch in neuroblastoma. Notch pathway targets and effectors were highly induced in our RNA-seq of IMR32 treated with BMP4, so we first validated a panel of Notch pathway genes, including *HES1*, *MAFB*, *MAML2* and *NOTCH3* (Figure 7A), confirming a BMP4-Notch signaling pathway in neuroblastoma. We further examined BMP4 effects on Notch signaling by protein analysis of IMR32 and SK-N-BE(2)-C cell lysates. In both cell-lines, simultaneous induction of MSX1 and cleavage of NOTCH3 was observed, together with upregulation of the Notch target and effector protein HES1 (Figure 7B). Real-time PCR confirmed that BMP4 induced Notch genes in SK-N-BE(2)-C cells too (Figure 7C). Finally, we evaluated the regulation of NOTCH3-ICD induced genes identified in IMR32 cells [39] in our BMP4 and Wnt RNA-seq data. GSEA revealed a striking positive correlation between BMP4 and NOTCH-ICD induced genes, and similarly with BMP4 and NOTCH-ICD repressed genes (Figure 7D). Our Wnt DEGs also showed a correlation with the NOTCH3-ICD regulated transcriptome in neuroblastoma, but to a lesser extent than BMP4 (Figure 7E). Together these studies confirm the strong interplay of Wnt, BMP and Notch signaling in determining transcriptional programmes in neuroblastoma, leading to growth arrest and, in some cell-lines, differentiation (Figure 8).

## 4. Discussion

The molecular etiology of neuroblastoma has been intensively studied at the level of deregulated transcriptomes resulting from altered developmental transcription factor expression, as exemplified by MYCN. Perhaps surprisingly for an embryonic tumor associated with a disrupted differentiation programme, our understanding of the signaling pathway networks that are involved in neuroblastoma tumorigenesis remains relatively limited. In particular, the role of the Wnt signaling pathway, often clearly oncogenic in both adult and childhood cancers such as colorectal cancer and Wilms’ tumor [1], has been found to exert more complex influences in NB, as shown several laboratories, including ours. These data have indicated Wnt-induced growth promotion or suppression [13,16,40,41], modulation of signaling and transcriptional pathways, interactions with MYCN [16,41], and underlying changes in differentiation states contributing to neuroblastoma tumor heterogeneity [16,19]. In this study, we define a Wnt-BMP growth inhibitory axis in neuroblastoma, identifying MSX and Notch signaling as downstream mediators of growth suppression. 

Previous studies have suggested a role for BMP signaling in regulating neuroblastoma cell differentiation and proliferation. However, the upstream and downstream mechanistic intermediates of BMPs have never been characterized, including the crosstalk with other signaling pathways. BMP2 induced neural differentiation in mouse neuroblastoma cells by increasing the expression of neurogenic transcription factors *Dlx2*, *Brn3a*, and *NeuroD6* [22]. Similarly, BMP2 was shown to induce growth arrest and neuronal differentiation of SH-SY5Y and RTBM cell-lines subsequent to accumulation of p27 [23]. Although BMP2 has been shown to be a Wnt target gene in other cell types, our RNA-seq of Wnt3a/Rspo2 treated neuroblastoma cells highlighted BMP4 as the most prominent BMP gene downstream of Wnt/β-catenin signaling, and a recent study showed that BMP4 can trigger neurite-like extensions in SH-SY5Y cells, induce TrkA, and decrease MYCN and Ki-67 [24]. Our study in part agrees with these findings, but with important differences and clearer mechanistic insights. Our studies quantify and confirm growth suppression of not only SH-SY5Y cells, but also SK-N-BE(2)-C and IMR32 cells, together with increases of TrkA. Clear neuritogenesis was only observed in IMR32 cells. Furthermore, our studies do not suggest that reduced MYCN levels are the predominant effect of BMP4 treatment, Ferlemann et al. having suggested this based on MYCN reduction in the non-MNA neuroblastoma line SH-SY5Y [24]. Given our confirmation of G1-phase cell-cycle arrest, reductions in levels of MYCN and E2F1 may be attributable to cell-cycle effects. However, based on GSEA analysis of our RNA-seq data, which demonstrate a strong effect of BMP4 on MYCN-mediated gene regulation, we suggest that BMP4 might more subtly interfere with MYCN transcriptional activity.

Our transcriptomic analysis did not support activation of *DLX2*, *BRN3A*, or *NEUROD6* genes as being involved in the BMP4-induced phenotypes of neuroblastoma cells, as previously suggested by studies using BMP2 [22]. Rather, our studies confirmed strong and consistent induction of MSX proteins, in particular MSX1. MSX1 is already known to be downstream of BMP signaling in the developing neural crest [31] and has been shown to suppress proliferation and colony formation in soft agar when exogenously over-expressed in neuroblastoma cells. This study by Revet et al. further established that MSX1 activated NOTCH3 and that low mRNA expression of the Notch pathway target *HEY1* correlated with poor prognosis [18]. The suggested link between BMP, MSX and Notch signaling was verified by both our RNA-seq and protein expression analyses. The role(s) of Notch signaling in neuroblastoma are complex, and although recent studies have linked it primarily with tumor heterogeneity and differentiation states of neuroblastoma [21,42], other studies, in addition to Revet et al. [18], have suggested that Notch signaling can be growth suppressive in neuroblastoma. In particular, neuroblastoma growth inhibition was demonstrated following exogenous expression of the intracellular domain of all three NOTCH proteins (NOTCH1-3). Furthermore, transduction with HES1 of several neuroblastoma cell-lines, including IMR32 and SH-SY5Y, led to inhibition of proliferation, and growth was also inhibited by treatment with recombinant Notch ligand Jag1 [43]. Another study associating Notch signaling with cell migration reported that the NOTCH3-ICD led to a transient attenuation of the cell-cycle [39]. Importantly, a study demonstrating the therapeutic benefits of the histone deacetylase inhibitor panobinostat in vivo, using the Th-MYCN mouse model of neuroblastoma, demonstrated that growth inhibition and differentiation resulting from panobinostat were accompanied by increased Notch and BMP pathway genes [44]. On the basis of published work and our data, we propose that the net effect of BMP signaling in neuroblastoma is contingent on signaling crosstalk with Notch. 

Whilst our analyses provide, for the first time, strong evidence for the interplay of Wnt, BMP4 and Notch signaling in neuroblastoma, it is unlikely that the profound effects of BMP4 on the neuroblastoma transcriptome and phenotype are mediated solely by any factor, but are rather a co-ordinated effect of the complex interactions of signaling pathways and developmental transcription factors. BMP4 can also control growth by non-SMAD-dependent pathways such as MEK/ERK signaling [45] and it will be important to conduct phosphoproteomic analyses in the future. Nevertheless, our compelling demonstration of BMP4 protein absence in poorly differentiated and aggressive neuroblastomas, and BMP4’s strong anti-proliferative effect raise the possibility of exploiting BMP4 or agonists of BMP signaling as therapeutic agents. Interestingly, a BMP9-derived peptide has been shown to enhance the differentiation of neuroblastoma cells [46]. Our characterization of the Wnt-BMP-MSX-Notch pathway and its associated biomarkers rationalises the evaluation of such novel therapeutics, as well as providing a foundation for delineating signaling regulatory networks in neuroblastoma.

## Figures and Tables

**Figure 1 cells-09-00783-f001:**
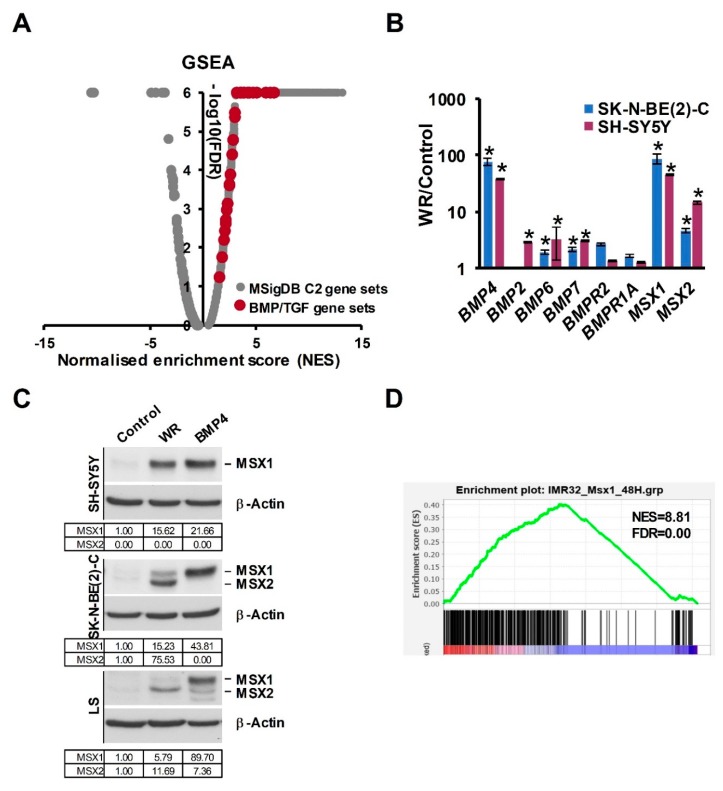
Convergence of Wnt and BMP pathways in NB. (**A**) GSEA Volcano plot on RNA-seq data of Wnt3a/Rspo2 (WR) induced SK-N-BE(2)-C cells (ERP023744) with C2 group of gene sets (Molecular Signatures Database, Broad Institute), highlighting upregulation of BMP and TGF-β sets, shown in red. (**B**) Genes coding for BMP ligands, receptors and BMP target genes *MSX1* and *MSX2* were upregulated by 72H Wnt3a/Rspo2 treatment in SK-N-BE(2)-C and SH-SY5Y cells as detected by qPCR. Statistically significant differences (*p* < 0.05) are indicated by asterisks (*n* = 3). (**C**) Western blot of neural crest master regulators MSX1 and MSX2 induction by both Wnt3a/Rspo2 and BMP4 treatment in neuroblastoma cells after 72–96H treatments. (representative of *n* = 2). Quantification relative to the control samples, after normalization to the loading controls, is shown in the tables. (**D**) GSEA showing upregulation of MSX1 target genes (GSE16481, 48H induction) in WR-treated SK-N-BE(2)-C.

**Figure 2 cells-09-00783-f002:**
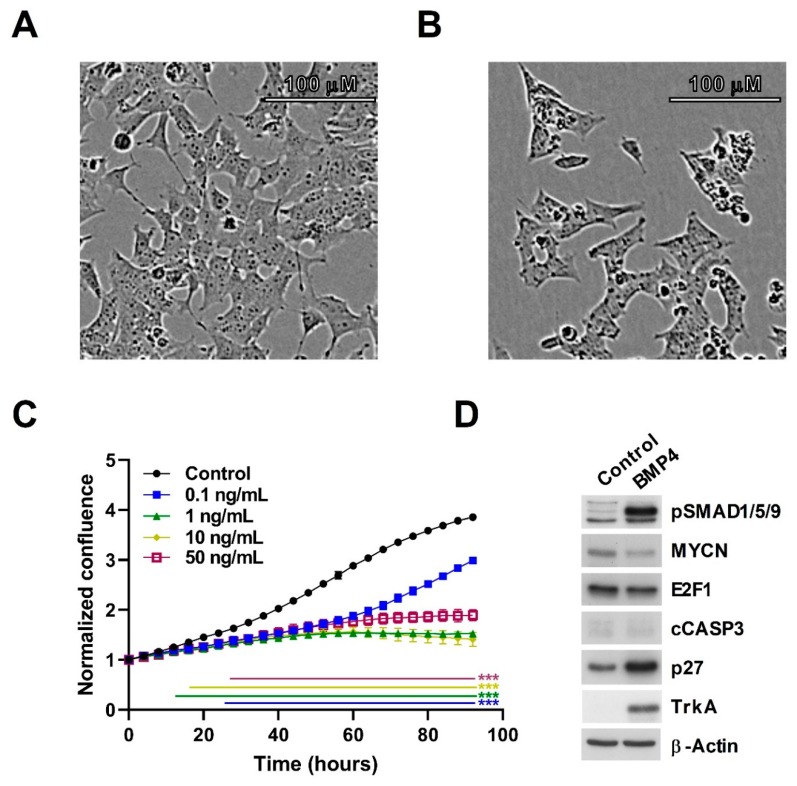
BMP4 treatment induces growth inhibition in SK-N-BE(2)-C cells. (**A**) Phase contrast image of vehicle-treated SK-N-BE(2)-C cells, and (**B**) 10 ng/mL BMP4-treated SK-N-BE(2)-C cells after 96 h. (**C**) Cell confluence was measured and analysed in triplicates by Incucyte live cell imaging and was used as a surrogate for growth. Normalized confluence was significantly (*p* < 0.01) reduced by BMP4 in all concentrations tested by 30 h after treatment, based on analysis of triplicate treatments. This experiment is a representative of 3 biological replicates (*n* = 3). (**D**) Western blot of changes in protein expression and phosphorylation after 96 h of BMP4 treatment. Quantification of Western blot data is shown in Appendix A.

**Figure 3 cells-09-00783-f003:**
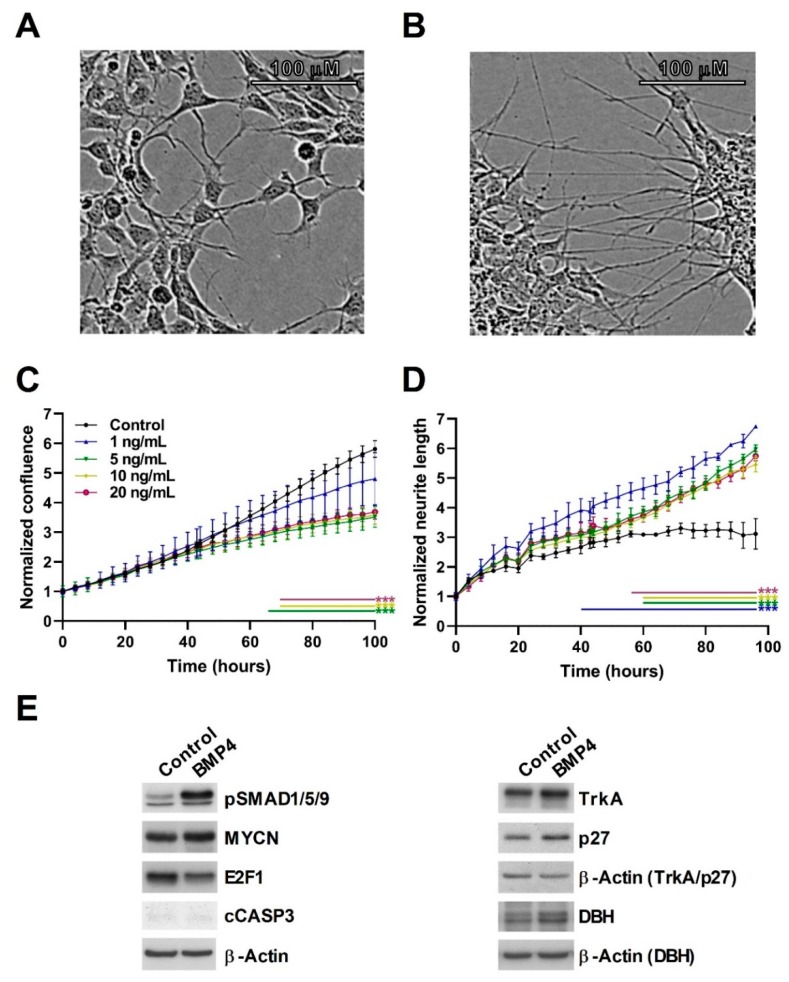
BMP4 treatment induces growth inhibition and phenotypic change in IMR32. Phase contrast image of (**A**) vehicle and (**B**) BMP4-treated IMR32 cells after 96 h (10 ng/mL). (**C**) Normalized cell confluence, measured in triplicates, was significantly (*p* < 0.01) reduced by BMP4 in concentrations above 5 ng/mL (*n* = 3). (**D**) Significant changes (*p* < 0.01) in normalized neurite length was observed after treatment with BMP4 at concentrations above 1 ng/mL (*n* = 3). (**E**) Western blots showing protein expression and phosphorylation changes after 48–72 h BMP4 treatment.

**Figure 4 cells-09-00783-f004:**
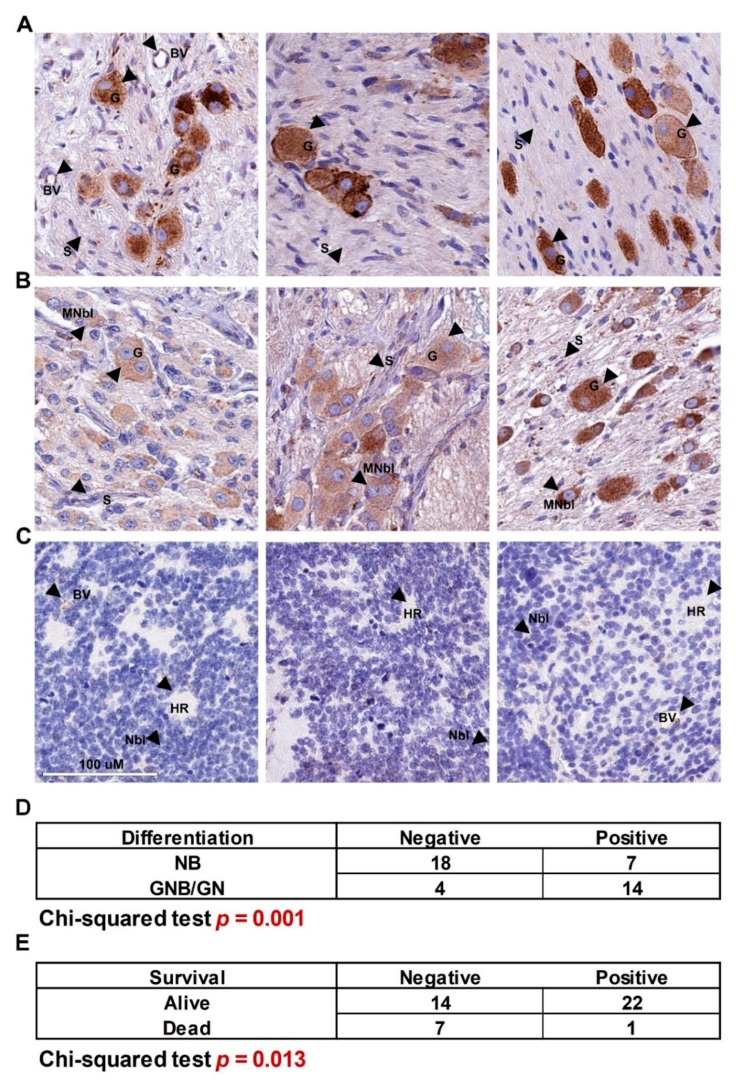
BMP4 immunohistochemistry in neuroblastic tumors. Expression of BMP4 protein in (**A**) mature ganglioneuroma, (**B**) ganglioneuroblastoma and maturing ganglioneuroma, (**C**) poorly differentiated neuroblastoma, where the third tumor is *MYCN*-amplified. BMP4 expression was positively and significantly correlated to (**D**) the degree of differentiation and (**E**) survival (Chi-squared test). Neuroblasts (Nbl), maturing neuroblasts (MNbl), ganglion cells (G), Schwannian stroma (S), Homer Wright rosettes (HR) and positively staining blood vessels (BV) are indicated with arrows.

**Figure 5 cells-09-00783-f005:**
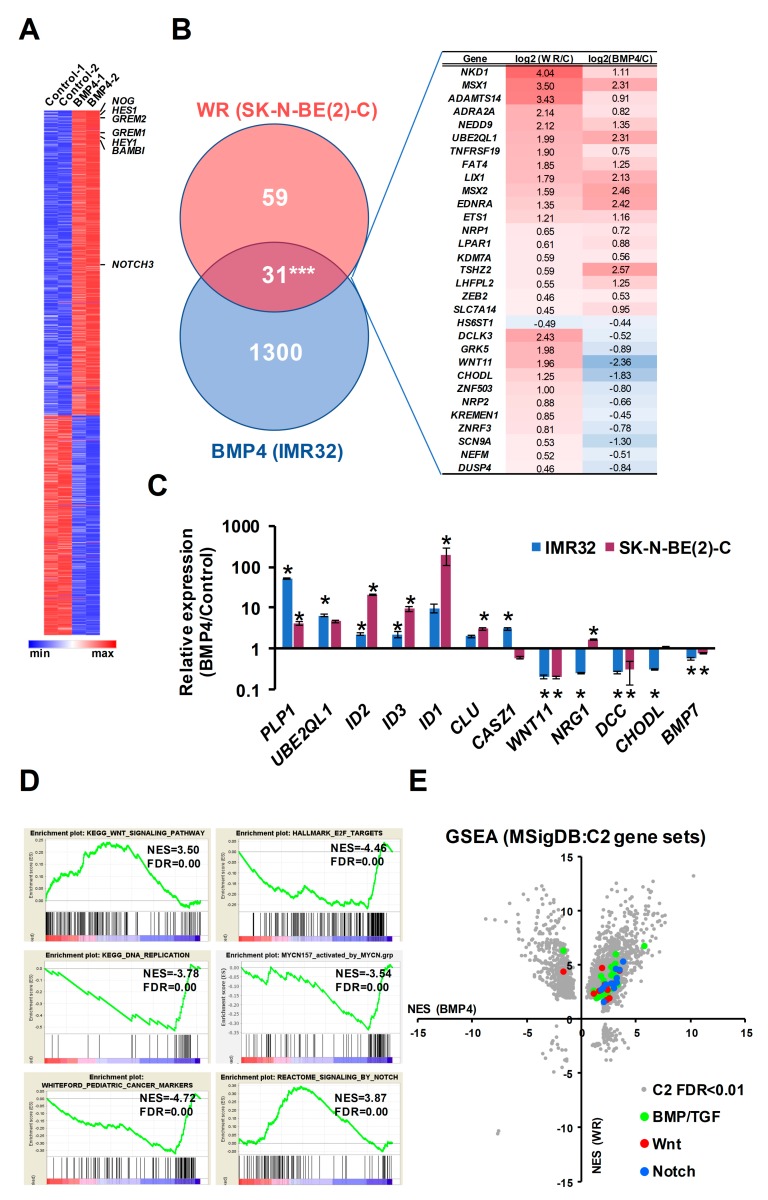
RNA-seq analysis of BMP4-treated IMR32 cells reveals Wnt and BMP4 signaling crosstalk in neuroblastoma cells. (**A**) Heatmap of DEGs in IMR32 after 24H BMP4 treatment (5 ng/mL), *p* < 0.005 (DESEQ2), minimum fold change 1.3. Canonical BMP4 and Notch target genes are indicated. (**B**) Venn diagram and heatmap of shared target genes of BMP4 and WR datasets in neuroblastoma cells. The number of shared genes is significantly higher than expected by chance (hypergeometric test *p* = 4.8e−22). (**C**) Validation of BMP4 target genes in 2 neuroblastoma cell lines; statistically significant differences (*p* < 0.05) are indicated by asterisks (*n* = 3). (**D**) GSEA highlighting significant regulation of functional gene sets. (**E**) Comparative GSEA analysis of C2 gene sets (Molecular Signatures Database, Broad Institute) in BMP4-treated IMR32 and WR-treated SK-N-BE(2)-C cells reveals upregulation of BMP/TGF (*n* = 29), Wnt (*n* = 12) and NOTCH (*n* = 11) gene sets by both treatments.

**Figure 6 cells-09-00783-f006:**
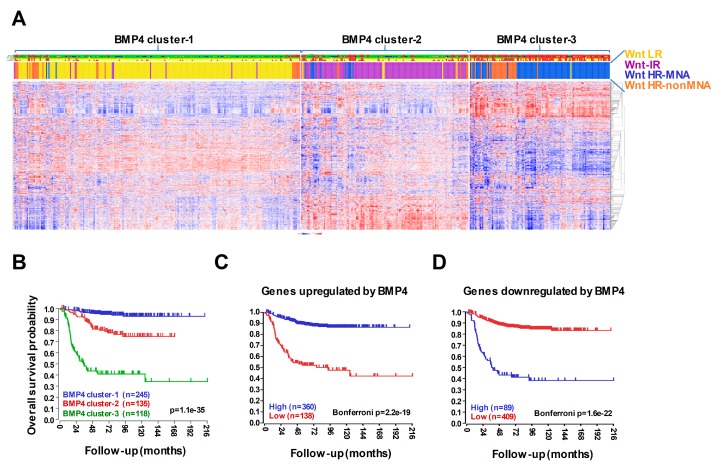
Meta-analysis of BMP4 target gene expression in primary neuroblastomas. (**A**) K-means clustering of BMP4 target genes in primary tumor gene expression data set SEQC (*n* = 498, k = 3). The coloured bars on the top indicate risk status, stage, *MYCN* amplification, survival, progression and clustering according to WR target genes. (**B**) Clustering according to BMP4 target gene expression divided the SEQC patient cohort into prognostic groups with significantly different survival probabilities. (**C**) Kaplan–Meier analyses showing that high expression of BMP4-upregulated genes strongly and significantly correlates with survival, while (**D**) high levels of genes downregulated by BMP4 is associated with poor prognosis.

**Figure 7 cells-09-00783-f007:**
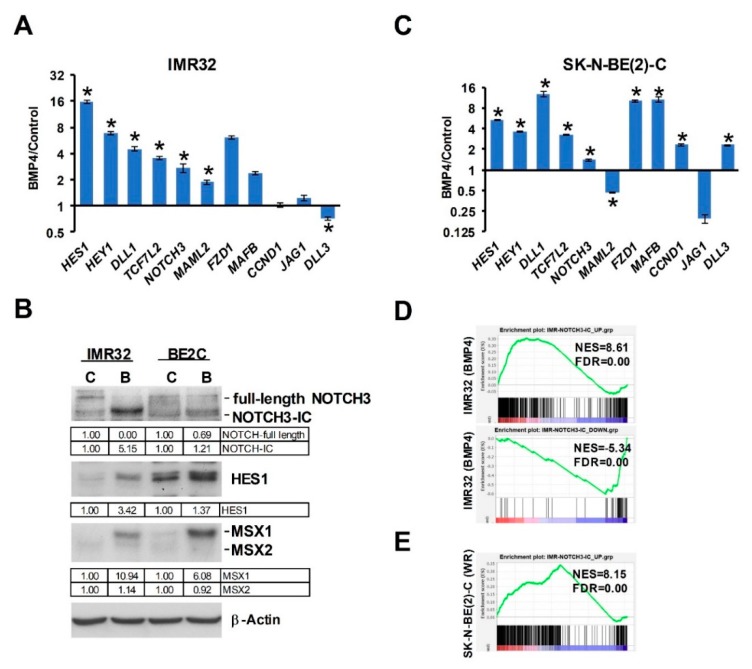
BMP4 and Wnt regulate Notch signaling in neuroblastoma. (**A**) Regulation of Notch pathway genes by BMP4 in IMR32 cells, detected by qPCR. Statistically significant differences (*p* < 0.05) are indicated by asterisks (*n* = 3). (**B)** BMP4 treatment leads to NOTCH3 cleavage, upregulation of Notch target/effector protein HES1 and MSX1/2 proteins in IMR32 and SK-N-BE(2)-C cells. (**C**) Regulation of Notch pathway genes by BMP4 in SK-N-BE(2)-C cells, detected by qPCR (*n* = 3). (**D**) GSEA of RNA-seq data sets of BMP4-treated IMR32 and (**E**) Wnt3a/Rspo2-induced SK-N-BE(2)-C with NOTCH3-IC target gene sets identified in IMR32.

**Figure 8 cells-09-00783-f008:**
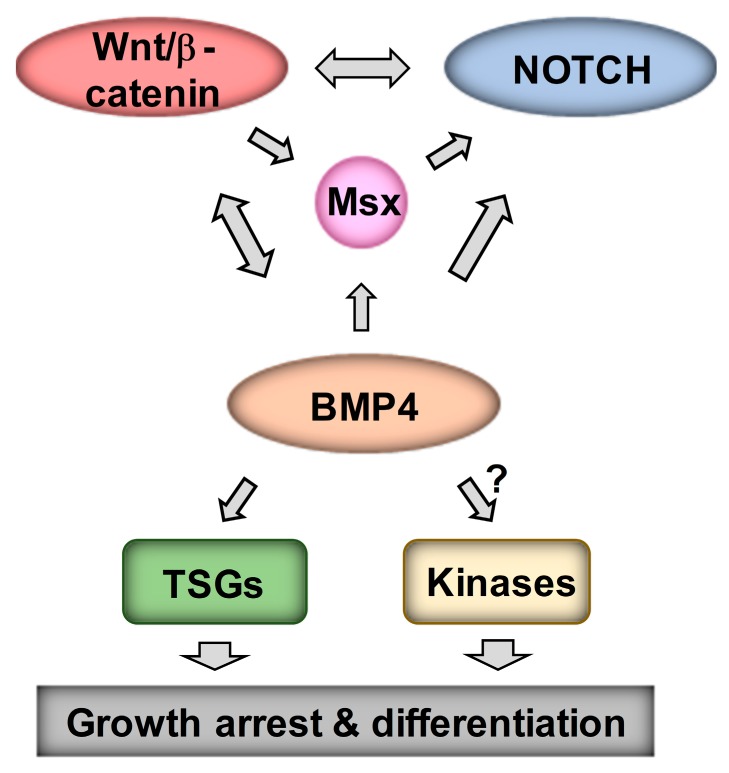
Model for Wnt, BMP, MSX and Notch crosstalk. Regulatory interactions promoting growth arrest and differentiation in neuroblastoma are outlined.

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
