# Peer review of "A Wnt-BMP4 Signaling Axis Induces MSX and NOTCH Proteins and Promotes Growth Suppression and Differentiation in Neuroblastoma"

_cells, 2020, doi:10.3390/cells9030783_

Round 1
Reviewer 1 Report
In this study, the authors show the relevance of BMP4 in growth suppression and differentiation of neuroblastoma cell lines. Moreover, by immunoistochemistry in neuroblastic tumors, they get more evidence of an oncosuppressive role of BMP4. RNA-seq experiments, performed by the authors, demonstrate that the crosstalk of Wnt, BMP, and Notch signaling can be relevant in the suppression of neuroblastoma tumor growth.
The study is solid and the experiments are presented in a clear manner.
I have just minor suggestions.
1) The authors could add a brief sub-section “Statistical analysis” in “Materials and Methods” to indicate all statistical tests used;
2) at lines 183-184 authors state that other cell lines were tested and a similar growth suppressive effect was found. I suggest to delete these lines since the experiments performed in the cell lines tested (SK-N-BE(2)-C, IMR32 and SH-SY5Y) are convincing. It is better to avoid the sentence “data not shown” because other results can be put in the section of supplementary results.
Author Response
First of all, we would like to thank the reviewer for the positive review and the helpful suggestions, which have greatly helped improving the manuscript.
In response to the specific points raised, (1) we have added a section describing all statistical methods used, as requested (Section 2.7 at line 178).
(2) We have also deleted the sentence referring to “data not shown” (line 251).
Reviewer 2 Report
The study by Szemes et al is a well-described and conducted study that used several techniques to address an important question regarding Wnt pathway and the role of BMP4 in neuroblastoma.
In the next lines, I provide some comments to further improve the quality of the manuscript:
- Since it is an open access journal with no space restriction, the authors must briefly describe all experimental-related information. Attention to items 2.2, 2.3, 2.4, 2.6. I encourage the authors to avoid reference a previous study to describe a technique that was used in the manuscript.
- Please describe the analysis method used in qPCR experiments and which normalizer was used.
- Several pictures are blurred. Please provide high-quality pictures.
- Provide a quantitative analysis for all western blot data in the manuscript.
- Provide all certification number of ethics committee for Tissue microarray data.
- In some legends there is no mention of N number. Please verify
- Why have the authors not evaluated the TCGA database? This reviewer thinks that the authors should evaluate the main findings of this study in the TCGA Neuroblastoma database.
- The authors use PI staining to evaluate all phases of cell cycle, which is very limited since you cannot distinguish properly between S and G2/M. For that analysis the authors should use PI and Brdu staining. In addition, this reviewer does not see a cell cycle arrest since the other phases can be clearly seen. I would rather say a reduction in cells entering the cell cycle, with an increased in G1 population.
Author Response
The study by Szemes et al is a well-described and conducted study that used several techniques to address an important question regarding Wnt pathway and the role of BMP4 in neuroblastoma.
In the next lines, I provide some comments to further improve the quality of the manuscript:
Reviewer 2
- Since it is an open access journal with no space restriction, the authors must briefly describe all experimental-related information. Attention to items 2.2, 2.3, 2.4, 2.6. I encourage the authors to avoid reference a previous study to describe a technique that was used in the manuscript.
- Please describe the analysis method used in qPCR experiments and which normalizer was used.
- Several pictures are blurred. Please provide high-quality pictures.
- Provide a quantitative analysis for all western blot data in the manuscript.
- Provide all certification number of ethics committee for Tissue microarray data.
- In some legends there is no mention of N number. Please verify
- Why have the authors not evaluated the TCGA database? This reviewer thinks that the authors should evaluate the main findings of this study in the TCGA Neuroblastoma database.
- The authors use PI staining to evaluate all phases of cell cycle, which is very limited since you cannot distinguish properly between S and G2/M. For that analysis the authors should use PI and Brdu staining. In addition, this reviewer does not see a cell cycle arrest since the other phases can be clearly seen. I would rather say a reduction in cells entering the cell cycle, with an increased in G1 population.
Our response:
We would like to thank the reviewer for the positive comments and the helpful suggestions, which have greatly improved the manuscript.
Please find our response below to the specific points raised:
Reviewer 2
- We have added detailed descriptions to the Materials and Methods as requested (section 2.2: lines 102-108, 110-116; section 2.3: lines 120-129; section 2.4: lines 131-143; section 2.6: lines 163-168).
- We have added this information to Materials and Methods section 2.4 (lines 139 – 143).
- We have improved the resolution of all pictures (300 dpi).
- All Western blot data have been quantified and included in figures 1C and 7B and supplementary figure S1.
- We have inserted this in Materials and Methods section 2.5 (lines 152-156).
- Number of replicates have been now provided (lines 364, 366, 379, 391, 392, 414, 433, 436 and 517).
- Thank you for this excellent suggestion. We have checked the mutations and copy number changes that may affect the function or expression of the BMP4 gene in TCGA, but none have been reported in neuroblastoma. The only RNA-seq data set available in TCGA is the TARGET-NBL set, which we analysed to verify our findings in this data set. We have analysed the TARGET-NBL data set on the R2 platform, where it is also available and which allowed us to query the association of BMP4-regulated metagenes, determined in this study, with clinical correlates in TARGET-NBL. We found that genes upregulated by BMP4 significantly correlated with 1. favourable histology, 2. Low Ki67 marker expression, indicating lower proliferation rate and 3. increased survival, which re-enforces our findings that BMP4 induces growth inhibition, is correlated with a more differentiated histology and better prognosis. Conversely, genes downregulated by BMP4 significantly correlated with unfavourable histology, high Ki67 marker expression and lower survival rate. We have included these analyses in the TARGET-NBL data set in Supplementary Figure S6.
- We have amended our interpretation of the cell cycle data accordingly (line 240).
We hope that we have satisfactorily addressed the Reviewer’s concerns and that the manuscript now meets the high standard required for publication.
Reviewer 3 Report
It is an interesting paper with quite a lot of work done. However, presentation of data in the Figures and legends are too brief and not clear enough. Few things must be added to improve quality of presentation of data.
1) In the methods, there isn't a section of Statistic. Does it mean no statistic test has been performed in any dataset? Please add back in the methods if there is one. I am also wondering whether any of the results in bar graphs (1B, 5C, 7A and 7C) is significant as some of them look like they are. It would be great to perform relevant statistic tests on these results.
2) Please indicate the number of replicates (n=) in legend 1B, 5C, 7A ad 7C.
3) Please put a scale bar or magnification for 1A and 1B
4) It is so hard to see the lines for 2C, 3C and 3D. The colours are also very similar. Please put asterisk to indicate the lines that are statistically significant.
5) The labelling in Figure 4A, B and C are not clear. Adding arrows to indicate the location of each structure will certainly help a lot.
Author Response
It is an interesting paper with quite a lot of work done. However, presentation of data in the Figures and legends are too brief and not clear enough. Few things must be added to improve quality of presentation of data.
Reviewer3
1) In the methods, there isn't a section of Statistic. Does it mean no statistic test has been performed in any dataset? Please add back in the methods if there is one. I am also wondering whether any of the results in bar graphs (1B, 5C, 7A and 7C) is significant as some of them look like they are. It would be great to perform relevant statistic tests on these results.
2) Please indicate the number of replicates (n=) in legend 1B, 5C, 7A ad 7C.
3) Please put a scale bar or magnification for 1A and 1B.
4) It is so hard to see the lines for 2C, 3C and 3D. The colours are also very similar. Please put asterisk to indicate the lines that are statistically significant.
5) The labelling in Figure 4A, B and C are not clear. Adding arrows to indicate the location of each structure will certainly help a lot.
Our response:
We would like to thank the reviewer for the appreciation of our work and the insightful comments, which helped us to improve the manuscript.
Please find our response below to the specific points raised:
1) We have added a Statistical analysis section to Materials and methods (Section 2.7, lines 178-195). We also included a statistical analysis of all qPCR data (Figures 1B, 5C, 7A and 7C). The figures show analysis of representative samples, and statistical significance is indicated by asterisks (p<0.05) based on analysis of biological replicates.
2) We included the number of replicates in the legends (lines 367, 412, 431 and 434).
3) Scale bars were added to all images of cells (Figures 2A, 2B, 3A,3B, S3A and S3B).
4) We have reduced the amount of data shown on these charts, without affecting the message, which allowed us to use more distinctive colours to make the graphs clearer. Statistical significance was assessed for every time points, therefore it is indicated at the bottom of the charts using asterisks (*** = p<0.01) and bars, which indicate the time points at which this level of significance was achieved.
5) Arrows have been added to Figure 4. to pinpoint the cells and histological structures.